# Electron-Beam Domain Patterning in $Sr_{0.61}Ba_{0.39}Nb_2O_6$ Crystals

**Tatyana R. Volk** [1,*], **Lyudmila S. Kokhanchik** [2], **Yadviga V. Bodnarchuk** [1], **Radmir V. Gainutdinov** [1], **Eugene B. Yakimov** [2] and **Lyudmila I. Ivleva** [3]

[1] Shubnikov Institute of Crystallography of FSRC "Crystallography and Photonics" RAS, Moscow 119333, Russia; deuten@mail.ru (Y.V.B.); rgaynutdinov@gmail.com (R.V.G.)
[2] Institute of microelectronics technology and high purity materials RAS, IMT RAS, Chernogolovka 142432, Russia; mlk@iptm.ru (L.S.K.); yakimov@iptm.ru (E.B.Y.)
[3] Prokhorov General Physics Institute of the RAS, Moscow 119991, Russia; ivleva@ran.gpi.ru
[*] Correspondence: volk@crys.ras.ru; Tel.: +7-499-135-6100

**Abstract:** The characteristics of electron-beam domain writing (EBDW) on the polar and nonpolar surfaces of the field-cooled (FC) and zero-field cooled (ZFC) $Sr_{0.61}Ba_{0.39}Nb_2O_6$ (SBN) crystals are presented in the range of accelerating voltage U from 10 to 25 kV. The exposure characteristics of the domain diameter $d$ and length $L_d$ (when writing on the polar and nonpolar surfaces, respectively) were measured. With increasing exposure time, $d$ tends to a saturation value, whereas $L_d$ grows linearly, the frontal velocity $V_f$ being of 40 μm/s. At U = 25 kV the achieved $d$ and $L_d$ are of 7 and 40 μm, respectively. The observed peculiar features of EBDW—specifically the domain widening with exposure times and the effect of the polarization state of the crystal on the domain stability—are accounted for by the relaxor features inherent to this material. The effects of electron-beam (EB) irradiation on the local hysteresis loops is evidence of a domain fixation.

**Keywords:** electron-beam irradiation; strontium barium niobate; ferroelectric domains

## 1. Introduction

Ferroelectric nano- and microdomain patterning is demanded by various practical applications, primarily by the development of quasi-phase matching optical-frequency conversion [1] and high-density non-volatile memory [2]. The well-known approaches to the domain engineering at the submicro- and nanometric scales are the domain writing by dc-fields applied to the tip of an atomic force microscope (AFM) or by local irradiation by an electron beam (EB). To develop these microscopic methods for the domain engineering, the mechanism of domain formation requires to be investigated. The mechanism of EB induced ferroelectric switching is studied poorly so far. The vast majority of these studies was performed in single-domain $LiNbO_3$ and $LiTaO_3$ crystals. Here we present the results of studies in the domain formation under electron-beam (EB) irradiation in ferroelectric solid solutions $Sr_{0.61}Ba_{0.39}Nb_2O_6$ (SBN).

Domains can be created by irradiation of either the polar or nonpolar crystal surfaces. In the former case (Figure 1a), a domain nucleated in the irradiation point grows axially into the crystal bulk. There is a sufficient number of publications on EB domain writing (EBDW) on the $LiNbO_3$ polar surfaces (for refs see, e.g., [2–5]). Under EB irradiation of the non-polar surface (Figure 1b), a domain nucleated in the irradiation point grows along the polar axis in a thin surface layer. The driving force of the lateral domain motion is the tangential component of the field induced by EB in the irradiation point. Recently, we performed detailed studies of EBDW on the nonpolar $LiNbO_3$ surfaces and proposed an approach to the domain formation; the results are summarized in [6]. At

the same time, the results of EBDW on the polar surface raise a lot of questions, since the achieved domain depths $T_d$ in LiNbO$_3$ are up to hundreds of microns (e.g., [3–5]), which by orders of magnitude exceed the electron penetration depths. Similarly, the depths of domains written by AFM-tip voltages in LiNbO$_3$ are inexplicably large; to account for this phenomenon, the mechanism of "ferroelectric breakdown" was proposed [7].

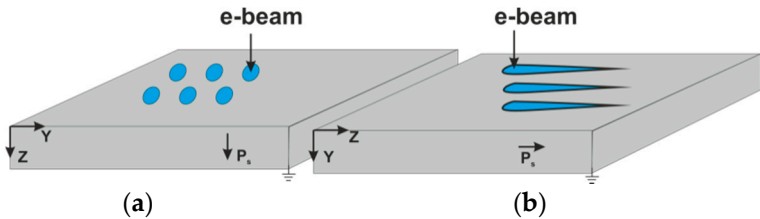

**Figure 1.** The scheme of electron-beam (EB) domain writing on the polar (**a**) and nonpolar (**b**) surfaces of a ferroelectric crystal.

In the given work, the EBDW was performed in the crystals of the congruent composition Sr$_{0.61}$Ba$_{0.39}$Nb$_2$O$_6$ (for brevity, denoted below as SBN). The motivation of these investigations is due to the fact that SBN in distinct from LiNbO$_3$ represents a convenient laboratory facility for studies in the ferroelectric aspect of EBDW. This is possible thanks to relatively low values of the phase transition temperature $T_c \approx 80\ °C$ and resulting relatively low coercive fields $E_c \approx (3\text{–}4)$ kV/cm. Extremely high $T_c$ and $E_c$ in LiNbO$_3$ prevent the investigations of this type. Specifically, in SBN a contribution from the domain state to EB-induced domain formation can be examined, whereas all EBDW experiments in LiNbO$_3$ and LiTaO$_3$ crystals were performed in single domain crystals. Additionally, SBN belongs to the uniaxial relaxor ferroelectrics, which promises certain specificity of EB induced domain formation.

Recently, the authors of [8] presented the studies in EBDW on the polar surface of SBN61-Ce crystals, covered by a photoresist layer. Using this method reduces essentially the irradiation dose required for EBDW [9,10].

In the given work, the results of studies in EBDW on the polar and nonpolar surfaces of SBN crystals are described. As will be seen, some effects observed in SBN—specifically the relaxation of written domains—differ fundamentally from those in LiNbO$_3$, which finds a qualitative explanation in the framework of current approach to polarization processes in relaxors.

To discuss the results obtained, a simplified scenario of the phenomena occurring under a local EB irradiation of insulators should be recalled following, e.g., [11–13]. In an irradiation point, a region of the negative space charge $Q_{sc}$ is accumulated as the result of capturing the primary electrons; the formed field $E_{sc}$ is responsible for the spontaneous polarization reversal under EB irradiation. The $Q_{sc}$ region is represented by a truncated sphere centered at the depth of $R_e/2$ (where $R_e$ is the primary electron penetration depth), whose value is determined by the EB energy and material properties. The kinetics of $Q_{sc}$ under given accelerating voltage U and its decay on EB turning- off are presented by the generalized expressions:

$$Q_{sc}(t) = Q_{sat}\left[1 - \exp\left(-\frac{t}{\tau_{eff}}\right)\right] \tag{1}$$

$$Q_{sc}(t) = Q_{sat}\exp\left(-\frac{t}{\tau_{eff}}\right) \tag{2}$$

where $Q_{sat}$ is the equilibrium negative space- charge value. The effective time constant $\tau_{eff} \approx 1/U$ is an empirical parameter depending on various factors such as the capture and backscattering of the primary electrons, electron trapping-detrapping, the radiation induced conductivity, etc.; $\tau_M = \varepsilon\varepsilon_0/G$ is the dielectric relaxation time, $\varepsilon\varepsilon_0$ and G are the dielectric constant and conductivity, respectively. The value of $Q_{sat}$ is governed by the total balance of currents:

$$I = I_Q + I_L + I_\sigma \tag{3}$$

where *I* is the incident EB current, $I_Q = dQ/dt$ is the displacement current related to electron trapping/detrapping, $I_L$ is the total leakage current. The simplified system of Equations (1)–(3) underlies the EB-induced processes of domain formation. Importantly, according to Equation (1) the space charge $Q_{sc}$ does not depend on the accelerating voltage U in the equilibrium state (since the exposure time $t_{irr} \gg \tau_{eff}$). (To be more precise, $Q_{sc}$ is slightly affected by U via the electron emission coefficient, depending on U). An equilibrium charge state in high-resistant dielectrics is attained during times from tens to hundreds ms [14]. As our exposure dependences were obtained in the range of $t_{irr}$ from 10 ms to seconds (see below), so, within a rough approximation we suggest the domain formation to occur under a quasi-equilibrium charge state.

Note, additionally to a negative space charge $Q_{sc}$ centered at a depth of $R_e/2$, a stable positively charged layer with a thickness of 3λ (where λ is the second-electron mean free path) is formed directly below the surface The occurrence of domains under EB irradiation of the positive polar surfaces of LiNbO$_3$ crystals [14,15] as well as the domain nucleation observed in the ferroelectric Rb-doped KTiOPO4 (RKTP)crystals placed in TEM [16] were accounted for by the polarization reversal in this layer

The paper presents the EBDW characteristics on the polar and non-polar SBN surfaces and a brief description of EB effects on the local piezoelectric hysteresis loops. The discussed domain sizes are the depth $T_d$ (along the EB propagation direction), the diameter *d* (when writing on the polar surfaces) and the length $L_d$ along the polar axis (when writing on the non-polar surfaces); the irradiation conditions are the accelerating voltage U of SEM, the EB current I, the inserted charge $Q = I_{tirr}$ and the irradiation dose $D = Q/S$ (where S is the irradiation area).

## 2. Materials and Methods

The congruently melting (SBN-0.61) crystals under study were grown by the modified Stepanov's technique [17] in the Institute of General Physics of the Russian Academy of Sciences (Moscow). The samples were 300–500 μm thick optically polished Z- and X- (Y-) cut plates. All samples were annealed at $T = 120\,°C > T_{max} = 81\,°C$, where $T_{max}$ is the temperature of the maximum dielectric permittivity in the region of the diffuse phase transition. On annealing, the samples were cooled down to room temperature either under an external field $E = 4$ kV/cm (so called field-cooled, FC-samples) or without fields (zero-field-cooling, ZFC).The domain writing was performed in a JSM-840A SEM equipped with the NanoMaker program which allowed us to control EB scanning over the surface and to evaluate the irradiation dose D. Experiments were carried out at SEM acceleration voltages *U* from 7 to 25 kV, EB currents $I = 1$–10 nA, the exposure times $t_{irr}$ from 5 ms to 10 s; the local irradiation area was $S = 0.5\ \mu m^2$; the irradiation doses $D = I \cdot t_{irr}/S$ were in the range from $10^4$ to $10^7\ \mu C/cm^2$. The current I was controlled with the aid of a Faraday cup located adjacent to the sample.

To estimate the values of $T_d$ corresponding to EB energies of 10 and 25 keV in SBN crystals, the spatial distribution of energy deposition rate as a function of depth were calculated using CASINO Monte Carlo code [18]. Of 1,000,000 electron trajectories were used in simulations. It is usually accepted that the excess carrier generated rate is proportional to the primary electron-beam energy deposition.

The domain sizes were determined by piezo-response force microscopy (PFM, Moscow, Russia). The domain diameter *d* (when EBDW on the polar surface) and the domain length $L_d$ (when writing on the non-polar surface) were measured using the axial and in-plane PFM modes, respectively. AFM experiments were carried out with a NTEGRA PRIMA AFM (NT-MDT SI, Moscow, Russia). Si probes with Pt conducting coating HA_FM/Pt (TipsNano, Tallin, Estonia) were utilized; the tip radius $R = 35$ nm, the cantilever stiffness $k \sim 3.5$ N/m and resonance frequency $f \sim 77$ kHz. To examine the domain configuration by PFM method, the electromechanical response $H_\omega$ was measured

$$H_\omega = \left[ \frac{1}{\kappa} \frac{dC}{dz} \left( \frac{V^\uparrow + V^\downarrow}{2} \right) \pm d_{ij} \right] U_{ac} \qquad (4)$$

where $d_{ij} = d_{33}$ and $d_{ij} = d_{15}$ are the piezoelectric coefficients for the axial and in-plane modes, respectively, $k$ is the force constant of the tip, $C$ is the tip-sample capacity, $\frac{V^\uparrow + V^\downarrow}{2}$ is the average contact potential difference between the tip and the crystal surface and $U_{ac}$ is the ac voltage between the tip and the electroded counter surface of the crystal. The sizes of the written domains were averaged over 5 points with the aid of SPIP 6.1.1 (Image Metrology, Lyngby, Denmark). The rms error was within 10%. Emphasize, the PFM scanning of EB written patterns was performed several hours after EB irradiation, so we could not detect the details (if any) of the domain decay during this time. As an auxiliary method, the low voltage SEM microscopy (denoted below as LVSEM, Chernogolovka, Russia) was used [19] which permitted us to examine the domain occurrence directly on the irradiation. The local piezoelectric hysteresis loops $H_\omega$-$U_{tip}$ were obtained in a pulse dc-mode in a voltage range +/-10 V with a step of 50 mV; the pulse durations $t_p$ were of 10 and 100 ms.

## 3. Experimental Results

Figure 1a,b depicts the schemes of domain formation under EB irradiation of the polar and nonpolar surfaces, respectively; Figure 2a–c presents the examples of phase PFM images of written domains. The PFM image of the background in Figure 2a reveals nanosized polar regions, which are characteristic for SBN. This nanoscale relief was interpreted as a manifestation of the relaxor features [20]. An elongated shape of domains is characteristic both for EBDW [6], (Figure 2b,c) and AFM-tip domain writing [21] on the non-polar crystal surfaces. However, as seen from Figure 2c, in SBN the shape of EB written domains becomes distorted with increasing $t_p$, which will be discussed below. The EB written domains both in SBN crystals under study and in SBN-Ce [8] can be reversibly thermally annealed by heating the sample to $T \approx 120\,°C > T_{max} = 81\,°C$.

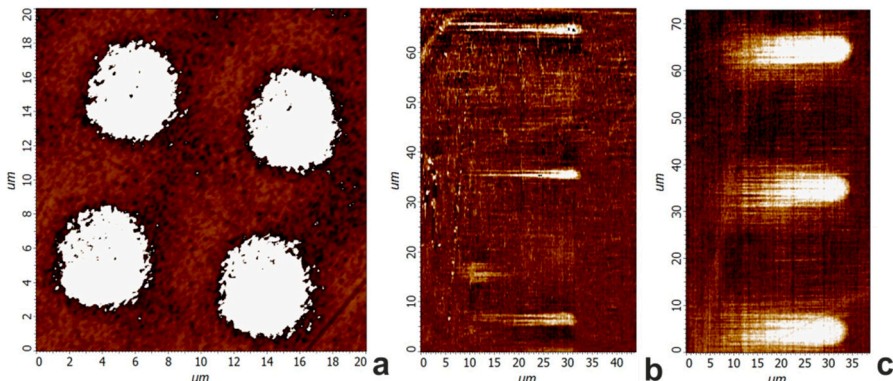

**Figure 2.** Piezo-response force microscopy (PFM) images of domains written on the polar (**a**) and nonpolar (**b**,**c**) $Sr_{0.61}Ba_{0.39}Nb_2O_6$(SBN) surfaces; the irradiation conditions are: U = 25 kV, I = 10 nA; $t_p$ = 500 ms (**a**,**c**) and 5 ms (**b**).

Regardless the crystal orientation, the doses D required for EBDW in SBN crystals by orders of magnitude exceed those in $LiNbO_3$, which is due to the drastically lower resistance in SBN ($\rho \approx 10^{-12}$–$10^{-13}$ [22]) as compared to $10^{-15}$–$10^{-16}$ $(\Omega\,cm)^{-1}$ in $LiNbO_3$ [2], thus to significantly higher leakage currents $I_L$.

Below, the results obtained in SBN crystals are compared to the EBDW data reported for $LiNbO_3$ of the congruent composition, which can be regarded as a reference material. It is necessary to emphasize that all EBDW results for $LiNbO_3$ crystals of any composition, presented in the literature, were obtained in single-domain crystals. For further consideration it is worth mentioning that all EB written domains in $LiNbO_3$ are completely stable regardless the geometry of writing.

The stable domains are known to arise under irradiation conditions exceeding certain "threshold" values ($U_{thr}$, $Q_{thr}$ and $D_{thr}$). When writing both on the polar and non-polar SBN crystal surfaces,

$U_{thr} = 7$ kV, which is close to $U_{thr}$ in LiNbO$_3$ [6]. As shown below, other conditions of the domain occurrence in SBN crystals are governed by the domain state and crystal orientation (Table 1).

**Table 1.** The $D_{thr}$ value for electron-beam domain writing.

| | $D_{thr}$, µC/cm$^2$ | |
| --- | --- | --- |
| | **Polar Surface** | **Non-Polar Surface** |
| LiNbO$_3$ | 100–1000 [3–5] | 500–1000 [6] |
| SBN | 10$^6$ | 10$^4$ * |

* The value obtained in ZFC crystal.

On irradiation of the polar SBN surface, the PFM scanning reveals the formed stable domains (Figure 2a) both in FC and ZFC crystals; the values of $D_{thr}$ are approximately the same. In contrast to this, the results of writing on the nonpolar surface are fundamentally anisotropic, since the PFM scanning performed after irradiation, detects the stable domains (Figure 2b,c),in ZFC crystals only. In FC crystals, the emerged domains are detected by LV SEM [19] directly on irradiation, however, the subsequent PFM scanning of the irradiated area finds no charge anomalies.

The $D_{thr}$ value for EBDW on the non-polar surface of ZFC crystals is by two orders lower than $D_{thr}$ required for writing on the polar surface (Table 1).

We now shortly describe the dependences of the domain sizes on the irradiation conditions. The depth $T_d$ in the irradiation point is evaluated based on $R_e$ at different accelerating voltages.

Figure 3a,b presents the spatial distribution of EB energy deposition rate as a function of crystal depth calculated using the CASINO Monte Carlo code [18]. The color lines show the relative distribution of the EB deposition in the EB range. Figure 3c shows the relative normalized profiles of the energy losses of irradiating electrons over the depth of SBN crystals at initial electron energies of 25 and 10 keV. As seen, the corresponding $R_e$ are of ~800 and 250 nm, respectively. According to the data on the absorbed energy, the main part of electrons is concentrated even at shallower depths from the surface ($T_d \approx 580$ nm and 140 nm for U = 25 kV and 10 kV, respectively) (Figure 3c).

Regrettably, it was impossible to compare the calculated $T_d$ to any experimental value, since the selective chemical etching [23] applied successfully for experimental evaluation of $T_d$ in LiNbO$_3$ and LiTaO$_3$ crystals (e.g., [6]), appeared to be ineffective in SBN.

In studies of EBDW on the nonpolar LiNbO$_3$ surfaces we have shown experimentally (for refs see [6]) that the domain depth $T_d$ in the irradiation point on the nonpolar (010) surface is equal to the primary electron penetration depth $R_e$. Since this result is in agreement with the general approach to the EB charging of insulators, we can extend our consideration to the case of SBN and to assume $T_d$ estimates performed above for this crystal ($T_d \approx 580$ nm and 140 nm for U = 25 kV and 10 kV, respectively) to be valid again for the domains written on the nonpolar surface. At the same time, the depth $T_d$ of domains written on the polar SBN surface "through" the photoresist layer was of tens of microns according to the confocal Raman microscopy measurements [8]. So, analogously to LiNbO$_3$ the depth of domains written on the polar surface of SBN crystals exceeds by orders of magnitude the electron penetration depth. This specificity of EBDW on the polar surface of ferroelectrics seems to be general and is worthy of further investigations.

Figure 4a,b presents the domain diameter $d$ on the polar surface and the domain length $L_d$ on the nonpolar surface vs. exposure time and dose $D$ (the dashed line in Figure 4b shows the linear approximation).

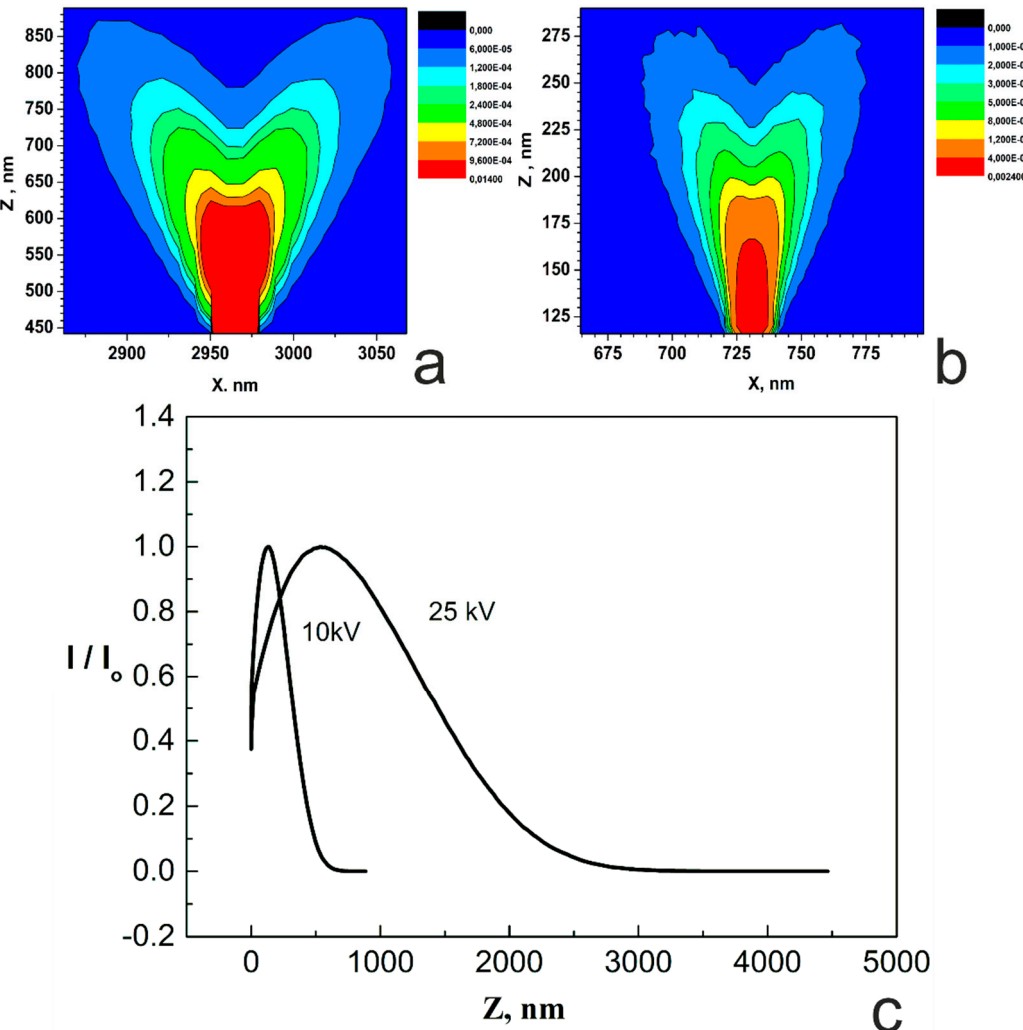

**Figure 3.** The spatial distribution of EB energy deposition rate (**a**,**b**) and the normalized profiles of electron energy losses over the depth (**c**) in SBN for 25 and 10 kV.

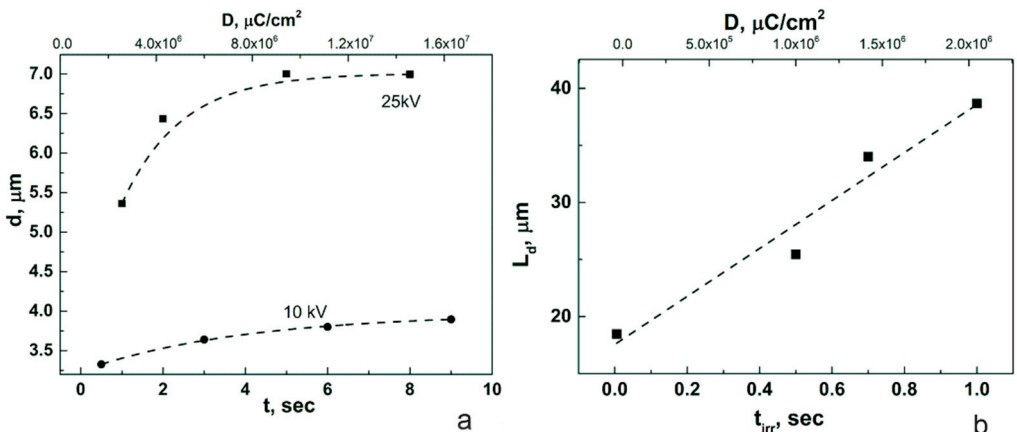

**Figure 4.** The diameter (**a**) and length (**b**) of domains, written on the polar and nonpolar surfaces, respectively, of zero-field cooled (ZFC) SBN crystals vs. exposure time and dose. In Figure 4b the dashed line shows the linear approximation; the results were obtained at U = 25 kV.

As shown below, the exposure characteristics presented in Figure 4 fit within the framework of the model description of ferroelectric switching [24].

We complete the presentation of experimental results by a short description of the EB effects on the local hysteresis loops $H_\omega$-$U_{tip}$. Figure 5 exemplifies the loops observed inside the irradiated area (Figure 5b) and in a close vicinity to it (Figure 5a) in ZFC crystal. In both cases, the loops were obtained at pulse durations $t_p$ = 10 and 100 ms; the loop characteristics are presented in Table 2. In the unirradiated area the values of coercive $U_c$ and bias $U_b$ voltages decrease noticeably with increasing $t_p$ (Figure 5b), which is expected a priori, since a decrease of $E_c$ and $E_b$ with increasing pulse duration was repeatedly observed in macro- and microscopic studies in the pulsed switching in SBN (e.g., [25,26]) and in another uniaxial relaxor PMN-0.4PT [27]. Oppositely, in the irradiated area of ZFC SBN the values of $U_b$ and $U_c$ depend negligibly on the pulse duration (Figure 5a, Table 2).

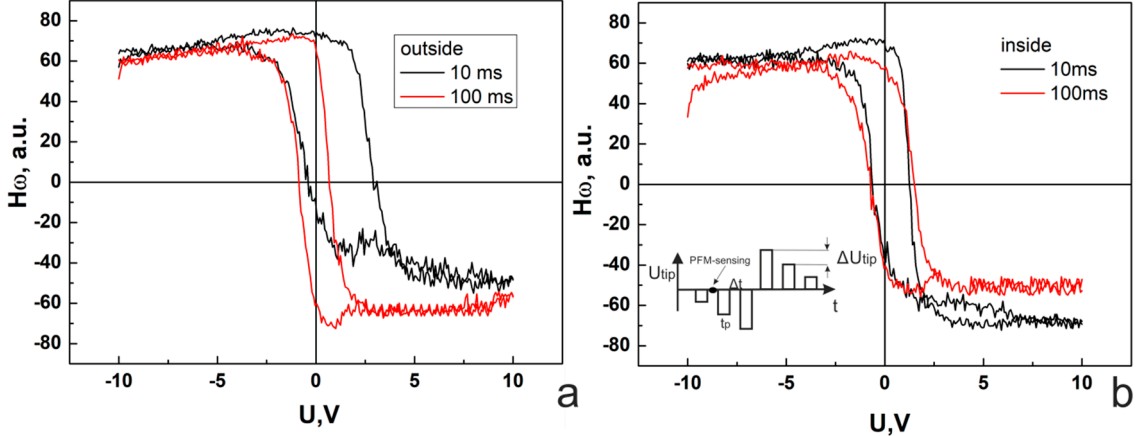

**Figure 5.** The local piezoelectric hysteresis loops measured in the closely spaced unirradiated (**a**) and irradiated (**b**) areas. The black and red loops correspond to the pulse duration $t_p$ = 10 and 100 ms, respectively. The inset shows the pulse train.

**Table 2.** The values of $U_b$ and $U_c$ in the irradiated area of ZFC SBN.

|  | Unirradiated Area | | Irradiated Area | |
|---|---|---|---|---|
| $t_p$, ms | 10 | 100 | 10 | 100 |
| $U_c$, V | 1.7 | 0.8 | 0.95 | 1.08 |
| $U_b$, V | 1.3 | 0.06 | 0.3 | 0.4 |

## 4. Discussion

We start the discussion from the exposure characteristics of the domain sizes. The domain diameter $d$ seems to come to saturation with D (Figure 4a). For comparison, at EBDW in SBN–Ce, the dependence of the domain area S on Q was controlled by the interdomain distance [8], namely, for the spatially spaced domains S(Q) was linear, whereas for the closely spaced ones it tended to saturation. This result was accounted for by an electrostatic repulsion between the closely spaced charged domain walls. A tendency of d(D) to come to saturation seen from Figure 4a can be related to the same reason, since the interdomain distance in our experiments was of 4–5μm.

The exposure dependence $L_d(t_{irr})$ on the nonpolar surface of ZFC crystal is shown in Figure 4b (as mentioned above, the domains written on the nonpolar surface of FC crystals were unstable). The dependence $L_d(t_{irr})$ is fitted by a linear function (Figure 4b). The linear dependences $L_d(t_{irr})$ were observed at EBDW on the nonpolar surfaces of LiNbO$_3$ and He-implanted optical waveguides on the nonpolar LiNbO$_3$ surface [6]. As mentioned in the Introduction, in the rough approximation the domain frontal motion in our case occurs under a quasi-equilibrium charge state, i.e., under a constant

space charge field. So, following [28] the linearity of $L_d(t_{irr})$ can be interpreted as a manifestationof the domain frontal growth by the viscous friction law

$$V_f \sim \mu E \qquad (5)$$

where $V_f = dL_d/dt_{irr}$ is the average velocity of domain frontal growth, $\mu$ is the domain-wall frontal mobility and $E$ is the driving field. This law describes the domain frontal motion under high external fields [24]. The estimate based on $L_d(t_{irr})$ (Figure 4b) gives $V_f \approx 40\mu m/s$, which is noticeably lesser than $V_f \approx 130 \mu m/s$ on the nonpolar LiNbO$_3$ surface observed under similar irradiation conditions [6]. A lower $V_f$ in SBN is obviously related to the domain-wall pinning effects characteristic for the uniaxial relaxors. The same order of magnitude of these velocities supports our assumption that the domain motion occurred under a stationary (saturated) $E_{sc}$. Summing up, the exposure dependences of the domain sizes (of the domain diameter $d$ and length $L_d$ on the polar and nonpolar surfaces, respectively) are qualitatively similar to those obtained in LiNbO$_3$. Unlike that, other EBDW characteristics, specifically found on the nonpolar SBN surface are fundamentally different from those in LiNbO$_3$. As shown below, it can be qualitatively accounted for by the relaxor features of SBN crystals.

According to the PFM images of domains written on the non-polar surfaces (Figure 2b,c), the domain shape is distorted as $t_p$ increases, namely, the domains become thicker and split. In particular, this distortion prevented us from extension of the $L_d(t_p)$ plots to larger $t_p$ (Figure 4b).

As known [11–13], the area of the space charge $Q_{sc}$ in insulators is widening with increasing exposure times. The domain thickening in SBN results from this effect and is due to a specificity of switching in the uniaxial relaxor ferroelectrics. Unlike in model ferroelectrics [24], in relaxor ferroelectrics the coercive field $E_c$ is characterized by a distribution function $D(E_c)$,so the switching is described by a wide spectrum of switching times [29]. With increasing $t_p$, the fraction of $P_s$ involved in the switching grows, which manifests itself in a domain thickening. The appearance of spatially resolved domain "tails" at large $t_p$ (Figure 2c) is caused by a spatial nonuniformity of $E_c$, which in its turn is due to a structural inhomogeneity of SBN crystals. The whole scenario differs fundamentally from that in the model ferroelectrics [24], LiNbO$_3$ among them, in which the switching is described by a uniquely determined coercive field $E_c$. In spite of the widening $Q_{sc}$ area, the area in which the $P_s$ reversal occurs is strictly confined by the condition $E_{sc} \geq E_c$. As a result, the needle-like shape of EB written domains in LiNbO$_3$ is not distorted and the length-to-width ratio is preserved up to highest $t_p$ of minutes [28].

Another distinction between EBDW in SBN and LiNbO$_3$ is manifested in the dependence of the stability of domains written on the nonpolar SBN surfaces on the polarization state of the crystal. The domains written in ZFC crystals are stable in the real time (Figure 2b,c). This is not at all the case in FC crystals. According to LVSEM measurements, in FC crystals, the domains do appear on irradiation, but quickly disappear since the PFM scanning does not reveal them. This specificity of the EB-written domain can be again attributed to the relaxor features of SBN for the following reasons: In contrast with model ferroelectrics, in which switching is controlled by domain nucleation [24], the switching of uniaxial relaxor ferroelectrics is governed by domain motion (e.g., [30–32]),which, in its turn, depends on domain wall pinning by inhomogeneities of a bulk internal field $E_{int}$, inherent to the uniaxial relaxors. Ferroelectric switching is governed by domain wall depinning, so in relaxor ferroelectrics, it is strongly hampered, compared to model ferroelectrics. In other words, to avoid the backswitching caused by pinning, the field pulses needed for the $P_s$ reversal in relaxor ferroelectrics should be by orders of magnitude longer than in model ferroelectrics. For example, when AFM domain writing in ZFC SBN crystals, backswitching times as long as one second were found [26], while those reported for BaTiO$_3$ and triglycine-sulphate(TGS) were of microseconds and milliseconds, respectively [24]. In FC crystals, the pinning effects are strengthened, which results, specifically, in an increased field pulse duration required for switching. Specifically, when macroscopic measurements of ferroelectric switching in SBN [33], the pulse durations needed to switch FC crystals were five times longer than those applied to ZFC ones (the field amplitude being the same).In the framework of this qualitative

consideration, in the described above EBDW experiments in SBN the used irradiation times $t_p \le$ one second were insufficient to create a stable reversed domain in a FC crystal. This reasoning accounts for a complete instability of a switched area detected in FC crystal directly on irradiation, on contrast to ZFC SBN crystal in which the domain written under the same irradiation conditions (Figure 2b,c) persisted in real time.

Let us briefly comment on the EB action on the hysteresis loop (Figure 5, Table 2). The main effect is an independence of $U_c$ and $U_b$ values on $t_p$ in the irradiated area. A decrease of $U_c$ and $U_b$ (i.e., of $E_c$ and $E_b$) with $t_p$, observed in the unirradiated area, is characteristic for the uniaxial ferroelectrics [25–27]. It was accounted for by a weakening of pinning under longer $t_p$ [31], which is qualitatively reasonable. In the framework of this speculations, an independence of $U_b$ and $U_c$ on $t_p$ in the irradiated area (Figure 5b) evidences indirectly of an increased domain-wall pinning, i.e., a domain stabilization after irradiation.

Ending the discussion, an enormous difference between $D_{thr}$ values of EBDW on the polar and nonpolar surfaces (Table 1) is unclear yet. It could be related to the fact that the domains written on the polar surface penetrate into the crystal bulk up to the depth of 80 micrometers [8]. The structural inhomogeneity and, additionally, an inferred domain fixation in the irradiation area, hamper the domain axial growth through the bulk. When writing on the nonpolar surface, the domain frontal motion in a thin surface layer of the order of 500 nanometers in thickness is facilitated as compared with the previous case.

## 5. Conclusions

In summary, a set of data on EBDW on the polar and nonpolar surfaces in FC and ZFC SBN crystals was obtained, which permitted us to outline the effects of EB irradiation on the ferroelectric properties of this material. The exposure dependences of the domain sizes (the domain length $L_d$ and domain diameter d when writing on the nonpolar and polar surfaces, respectively) are similar to those in $LiNbO_3$, so can be regarded as certain general regularity. At the same time, some characteristics of EBDW, chiefly obtained on the nonpolar SBN surface, differ markedly from those in $LiNbO_3$. In the first turn, these are a distortion of the domain shape with increasing irradiation time and a dependence of the writing scenario on the polarization state of the crystal, specifically, a complete instability of domains written in FC crystals. These phenomena find a reasonable qualitative explanation in the context of relaxor properties of SBN crystals.

**Author Contributions:** T.R.V. analyzed the data and wrote the paper; L.S.K. performed the experimental studies in EB irradiation; Y.V.B. and R.V.G. performed the AFM experiments and analyzed the data; E.B.Y. performed the calculations based on the M.C. method; L.I.I. provided the SBN samples. All authors have read and agreed to the published version of the manuscript.

**Funding:** This work was supported by the Ministry of Science and Higher Education within the state assignment, FSRC "Crystallography and Photonics" RAS and IMT RAS. The equipment of the Shared Research Center supported by the Ministry of Science and Higher Education [Project No. RFMEFI62114X0005]) was used in experiments. L.I. is indebted for support to Volkswagen Stiftung [Grant No. Az. 90.261].

**Conflicts of Interest:** The authors declare no conflict of interest.

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
