# Peer review of "Electron-Beam Domain Patterning in Sr0.61Ba0.39Nb2O6 Crystals"

_coatings, doi:10.3390/coatings10030299_

Round 1
Reviewer 1 Report
The manuscript by T. Volk et al. describe domain structure appeared in polar and non-polar cuts of Sr0.61Bs0.39Nb2O6 single crystals by electron beam domain writing. Authors observed domains by piezoresponse force microscopy method and analyze their size: diameter and length depending on the exposure time and radiation dose. Authors compared their data with those for LiNbO3 single crystals and attributed the difference to the relaxor character of the SBN crystals. The manuscript is well written. The presented results will be interesting for researchers working in field of ferroelectric materials. I have a few minor comments.
- For the non-polar cut authors showed data only for the radiation dose of 25 kV. What about 10 kV? Are the data qualitatively similar?
- In page 7 authors discuss the average velocity of domain frontal growth and say that the velocity for non-polar cut of SBN 40 microns/s is of the same order as that of LiNbO3 130 microns/s. In my opinion the difference is pretty large (more than three times) and should be discuss, maybe referring to the relaxor character of SBN.
Author Response
Dear Reviewer1,
We appreciate very highly your approval of our investigations and the valuable remarks
Now I respond to your comments
- For the non-polar cut authors showed data only for the radiation dose of 25 kV. What about 10 kV? Are the data qualitatively similar?
According to the preliminary results, the data for U = 10 kV are close to those for 25 kV. However, it needs to be clarified, so we decided not to include this material.
2. In page 7 authors discuss the average velocity of domain frontal growth and say that the velocity for non-polar cut of SBN 40 microns/s is of the same order as that of LiNbO3 130 microns/s. In my opinion the difference is pretty large (more than three times) and should be discuss, maybe referring to the relaxor character of SBN.
I agree with this comment. Now the text (lines 240 - 243) looks as follows
“The estimate based on Ld(tirr) (Fig. 4b) gives Vf μm/s, which is noticeably lesser than Vf μm/s on the nonpolar LiNbO3 surface under similar irradiation conditions [11,13]. A lower Vf in SBN is obviously related to the domain-wall pinning effects characteristic for the uniaxial relaxors. The same order of magnitude of these velocities supports our assumption that the domain motion occurred under a stationary (saturated) Esc”.
Reviewer 2 Report
This paper reports electron-bean domain patterning in SBN crystals.
This paper needs major revision to be accepted for publication.
- The domain area S on Q is discussed in Discussion 222-228. Have you experimented with different writing intervals? If so, add the result.
- It is desirable that the abstract include numerical information such as a domain growth rate and a domain size.
- Show why the origins of D and t are different on the X axis in Fig. 4.
Minor points are listed below.
- RKTP line 87, and TGS line 278 should be explained.
- Some parts are not subscripted. For example, lines 38, and Table 1. I would like to reconfirm.
Author Response
Dear Reviewer2,
We appreciate very highly your approval of our investigations and the valuable remarks.
Now I respond to your comments:
- The domain area S on Q is discussed in Discussion 222-228. Have you experimented with different writing intervals? If so, add the result.
We performed our measurements at the interdomain distance 4 – 5 µm only. When discussing a possible contribution from the interdomain distance, we refer to Chezganov e.a.[15],where this effect was observed.
- It is desirable that the abstract include numerical information such as a domain growth rate and a domain size.
We added to the Abstract the following remark “With increasing exposure time, d tends to a saturation value, whereas Ld grows linearly, the frontal velocity Vf being of 40 μm/s. At U = 25 kV the achieved d and Ld values are of 7 and 40 µm , respectively”.
- Show why the origins of D and t are different on the X axis in Fig. 4.
If I understood the question correctly, Reviewer asks why the time scales in Figs. 4 a and b are different. As mentioned in the text (lines 254-256), the measurements of the domain length Ld(tp) are limited by the range of relatively short tp since at higher tp the domains become thicker and split. At the same time, writing on the polar surfaces requires exposure times and doses by order of magnitude larger than on the non-polar one.
- RKTP line 87, and TGS line 278 should be explained
Done.
Round 2
Reviewer 2 Report
Appropriate revisions have been made. I think this manuscript can be accepted.
Author Response
Thank you.